# Understanding the Relationship between Prompts and Response Uncertainty in Large Language Models

## Abstract

Large language models (LLMs) are widely used in decision-making, but their reliability, especially in critical tasks like healthcare, is not well-established. Therefore, understanding how LLMs reason and make decisions is crucial for their safe deployment. This paper investigates how the uncertainty of responses generated by LLMs relates to the information provided in the input prompt. Leveraging the insight that LLMs learn to infer latent concepts during pretraining, we propose a prompt-response concept model that explains how LLMs generate responses and helps understand the relationship between prompts and response uncertainty. We show that the uncertainty decreases as the prompt's informativeness increases, similar to epistemic uncertainty. Our detailed experimental results on real-world datasets validate our proposed model.

## 1 Introduction

Large language models (LLMs) have demonstrated impressive performance across a variety of tasks (Google, 2023; OpenAI, 2023; Zhao et al., 2023). This success has led to their widespread adoption and significant involvement in various decision-making applications, such as healthcare (Karabacak & Margetis, 2023; Sallam, 2023; Yang et al., 2023), education (Xiao et al., 2023), finance (Wu et al., 2023b), and law (Zhang et al., 2023a). However, despite their rapid adoption, the reliability of LLMs in handling high-stakes tasks has yet to be demonstrated (Arkoudas, 2023; Huang et al., 2023a). The reliability is particularly critical in domains such as healthcare, where model responses can have immediate and significant impacts on human behavior and hence their well-being (Ji et al., 2023). Thus, understanding LLMs' reasoning and decision-making processes and how they influence response uncertainty is critical for their safe and reliable deployment.

To understand this importance, consider the mobile health (mHealth) application in which machine learning algorithms are integrated to monitor users' health conditions and provide advice on daily activities (Boursalie et al., 2018; Trella et al., 2022; 2023). Providing suggestions that can influence users' health is a form of intervention in human decision-making. For LLMs to be suitable for such use cases, they should be accurate and provide consistent intervention strategies, e.g., consider an LLM-powered mHealth app that suggests physical therapy (PT) routines to a patient recovering from surgery. mHealth ensures the patient adheres to their PT regimen during rehabilitation despite the discomfort it may cause. mHealth must provide not only good but consistent suggestions to encourage PT adherence. Any inconsistent behaviors from the app could undermine any progress made. Conversely, providing accurate and consistent responses helps make the system more reliable and trustworthy (Shin et al., 2022).

Traditionally, when model architecture is fixed, model improvement relies on better hyperparameters (Bischl et al., 2023), such as using a more suitable optimizer (Hassan et al., 2023), or train it with more/better data (Simon et al., 2023). Due to the emergent capability of LLMs (Dong et al., 2022), such as in-context learning (ICL) (Kojima et al., 2022) and chain-of-thought (CoT) prompting (Wei et al., 2022), it is possible to improve the model responses by guiding it with informative prompts having relevant instructions and exemplars. Through these techniques, the LLMs can more effectively use the relevant information acquired from the training data to generate better responses, even if the prompt itself does not directly reveal the ground truth (Liu et al., 2023; Sahoo et al., 2024).

The response generated by LLMs is a series of tokens sampled from probability vectors of tokens using various heuristics (Radford et al., 2018; 2019; Brown et al., 2020), such as beam search, nucleus sampling, and greedy decoding. Typically, tokens with higher probabilities are chosen sequentially to produce the final response. The response variations are controlled by LLM hyperparameters such as temperature ($T$), top-$k$, or top-$p$. While response variations benefit creative tasks like poem and essay writing, they can be detrimental for tasks requiring high reproducibility and consistency (Ganguli et al., 2022; Huang et al., 2023c). However, making LLMs generate deterministic responses is not ideal, as users preferences may vary (Wu et al., 2023a). Hence, a better approach is needed to understand the sources of response uncertainty and develop methods to reduce it naturally rather than masking it by adjusting LLM hyperparameters.

For a fixed or black-box LLM, response uncertainty can be controlled by two ways: adjusting LLM's hyperparameters (e.g., temperature) to control the randomness in generated response and providing more task-relevant information in input prompt (prompt informativeness). This paper focuses on the response uncertainty due to the input prompt while keeping the LLM hyperparameters fixed. Hence, we address the following question: ***How is the amount of relevant information about a task in the input prompt related to the uncertainty in the response generated by an LLM?*** We use the insight that LLMs implicitly learn to infer latent concepts during pretraining (Xie et al., 2022; Hahn & Goyal, 2023; Zhang et al., 2023b) and propose a novel *prompt-response concept (PRC) model* in Section 2.

Our PRC model conceptualizes how an LLM generates responses based on given prompts and helps understand the relationship between prompts and response uncertainty by measuring response uncertainty for prompts with varying information about the task. We provide theoretical results that show the *uncertainty of responses generated by an LLM decreases as the prompt informativeness and model quality increase*. We draw connection between the reducible response uncertainty and epistemic uncertainty, and using our PRC model, we *explain why adding more relevant information to the prompt to a better* trained LLM is a principled and effective method to reduce the response uncertainty. Finally, we corroborate the validity of the PRC model via experiments and provide a simulation for a healthcare use case to demonstrate its efficacy. Specifically, our contributions can be summarized as follows:

- In Section 2, we propose a prompt-response concept model to quantify the relationship between prompt informativeness and response uncertainty in LLMs.
- In Section 3, we prove that response uncertainty decreases with higher prompt informativeness and model quality. Using our PRC model, we relate the reducible uncertainty to epistemic uncertainty and explain how adding relevant information in prompt to a well-trained LLM reduces response uncertainty.
- Finally, we validate the PRC model through experiments and demonstrate its theoretical efficacy using different tasks derived from real-world datasets and a healthcare use case simulation in Section 4.

## 2 PROMPT-RESPONSE CONCEPT MODEL FOR LLM

In this section, we first define what we mean by *concept*. We then use the notion of concept to explain our proposed prompt-response concept model of LLM. Finally, we provide theoretical results that explain the relationship between prompt informativeness and the uncertainty of responses.

**Concept.** The definition of a concept varies across fields, e.g., in philosophy, a concept represents the fundamental unit of thought; in psychology, it is a mental construct; in linguistics, it refers to the semantic units that words or phrases represent; and in education, it denotes key ideas or principles. In this paper, we define the *concept* as an abstraction derived from specific instances or occurrences that share common characteristics (Fodor, 1998; Laurence & Margolis, 1999; Weiskopf, 2009; Wilmont et al., 2013). To understand the notion of concept, consider this example of the concept: *Species*, which includes a group of organisms that share common biological traits. Furthermore, a concept can be expressed as a sequence with semantic meaning (e.g., using natural language). Here, we use the term 'semantic meaning' (or semantically meaningful) to refer to something that conveys information that is understandable and extractable by humans (Hurford et al., 2007). Consider another example of a concept: the *personal bio*,[1] which consists of sentences giving information about a person's name, occupations, contributions, and other personal details.

---

[1] This example is adapted from *wiki bio* concept example given in Xie et al. (2022).

---

**Concept: *Personal bio* of Alan Turing**

Alan Turing was an English computer scientist, mathematician, and cryptanalyst. He introduced the Turing machine, which formalized the concepts of algorithms and computation, serving as a foundational model for general-purpose computers. Turing is regarded as the father of theoretical computer science. . . .

---

**Concept attributes.** At first glance, the exact set of attributes defining a concept is not immediately clear. For instance, whether a attribute like 'a certain hobby of Alan Turing' in the above example, is part of the concept is subjective. In practice, the attribute set of a concept depends on the data the LLM is trained on, e.g., if the hobby of Alan Turing appears in the training data, the LLM has access to this knowledge, incorporating it into the concept's attribute set. Additionally, the notions of 'concept' and 'attribute' are relative in nature. An attribute of sufficient complexity could itself be considered a concept, encompassing its own set of attributes. For illustration purposes, we use paragraph-level concepts and sentence-level attributes, instead of word- or token-level patterns, as these higher-level abstractions allow a better understanding of the relationships between different sentences within the text (Bates, 1995; Bogatyrev & Samodurov, 2016; Wang et al., 2024).

In the above example of *personal bio* concept, explaining a concept often involves multiple sentences, each contributing to a specific and meaningful facet of information about the concept (Piccinini & Scott, 2006). We refer to the information of each facet as a *concept attribute*, e.g., the sentence, "Alan Turing was an English computer scientist, mathematician, and cryptanalyst" provides information about the name, nationality, and occupation of Alan Turing. Thus, a concept can be fully characterized by all its attributes. Recent works by Gao et al. (2024), Lieberum et al. (2024), and Templeton (2024) provide mechanistic interpretability evidence suggesting that LLMs can learn concept-like features. In this work, we formalize the *definition of concept* and then use it to provide theoretical insights into the relationship between prompt informativeness and LLM response uncertainty.

## 2.1 PROMPT-RESPONSE CONCEPT MODEL

We aim to understand how the input prompt informativeness is related to the uncertainty in the responses generated by an LLM. To do so, we first introduce notations representing different variables used in this section. Let $\mathcal{X}$ denote the set of all prompts and $\mathcal{Y}$ denote the set of all responses generated by an LLM $f$, where $f : \mathcal{X} \to \mathcal{Y}$. Let $\mathcal{V}$ be the vocabulary containing all unique tokens. For any prompt $x$ and response $y$, we have $x \in \mathcal{V}^{|x|}$ and $y \in \mathcal{V}^{|y|}$, where $|\cdot|$ returns the number of tokens in prompt/response. For a given prompt $x \in \mathcal{X}$, the LLM $f$ generates a response $y \in \mathcal{Y}$ such that $y = f(x)$. Since the response $y$ can vary each time the LLM generates it, we can control these response variations using two main factors: prompt informativeness and LLM's hyperparameters, such as temperature $T$, top-$k$, or top-$p$, which control the randomness in the generated tokens.

This paper focuses solely on the latter aspect while keeping the LLM's hyperparameters fixed. Building on the earlier works' interpretation that LLMs implicitly learn to infer latent concepts during pretraining (Xie et al., 2022; Hahn & Goyal, 2023; Zhang et al., 2023b), we propose the prompt-response concept (PRC) model of LLM. This model conceptualizes how an LLM generates a response for a given prompt, which will be used to understand the relationship between prompts and the response uncertainty by measuring response uncertainty for prompts with varying information. The PRC model has three main components (as shown in Fig. 1): prompt concept, response concept, and mapping functions, whose details are given as follows.

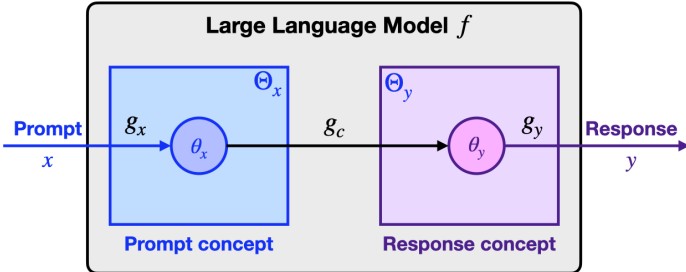

Figure 1: Prompt-Response concept model of LLM.

**Prompt concept.** Let $\Theta_x$ be the set of all concepts corresponding to prompts in set $\mathcal{X}$. We assume that each input prompt $x \in \mathcal{X}$ corresponds to a concept. We refer to this concept as the *prompt concept* $\theta_x \in \Theta_x$. Intuitively, an LLM recognizes input tokens as semantically meaningful units that

coherently describe an attribute of some latent prompt concept. The concept's attributes are expressed through multiple semantically meaningful sentences. If multiple sentences in the prompt cannot be combined to describe a single concept, the LLM treats them as representing different concepts. Our experiments in Fig. 4c show that *adding semantically meaningful information from different concepts can increase response uncertainty*.

**Response concept.** Let $\Theta_y$ be the set of all concepts corresponding to the possible responses in the set $\mathcal{Y}$. We refer to these concepts as the *response concept*. The PRC model assumes that each response concept $\theta_y \in \Theta_y$ is associated with a specific response $y \in \mathcal{Y}$.

**Mapping functions.** To understand the relationship between input prompt, intermediate concepts (i.e., prompt and response concept), and response, we assume the LLM $f$ is a composition of three mapping functions: prompt-concept mapping function ($g_x$), concept-concept mapping function ($g_c$), and concept-response mapping function ($g_y$). Hence, we can represent the response as $y = f(x) = g_y(g_c(g_x(x)))$, where the function $g_x$ maps the input prompt to a prompt concept, the function $g_c$ maps the prompt concept to a response concept, and finally, the function $g_y$ maps the response concept to a response. For a well-pretrained LLM, its $g_x$ can accurately find all the attributes in the given input prompt; for a well-aligned LLM, its $g_c$ can accurately map from some $\theta_x$ to the most relevant $\theta_y$.[2] Therefore, we expect a better LLM to generate responses with lower uncertainty and higher qualities, as demonstrated by our experiments in Fig. 2a and Fig. 2c.

To generate a response, an LLM maps the input prompt to a prompt concept, which is then mapped to a corresponding response concept. Finally, the LLM uses the response concept to generate the final response. When a prompt lacks sufficient task-related information (i.e., it is less informative) due to attributes being under-specified, we can expect higher variability in the responses generated by the LLM because there potentially exists multiple concepts that satisfy the attributes found in the input prompt, as corroborated by our experimental results in Fig. 5a. To further understand this relationship, we will next formalize how the informativeness of prompts is related to response uncertainty.

## 2.2 RELATIONSHIP BETWEEN PROMPTS AND RESPONSE UNCERTAINTY IN LLMS

Let $\mathcal{X}_{\theta_x} \subset \mathcal{X}$ be the set of prompts with the same semantic meaning[3] and contain all information of the prompt concept $\theta_x$. Let $\mathcal{X}_s \subset \mathcal{X}_{\theta_x}$ be the set of prompts with the same semantic meaning $s$ and only contain partial information about the prompt concept $\theta_x$. Let $\mathcal{A}_\theta = \{a_{\theta,1}, \ldots, a_{\theta,m}\}$ represent the set of all the attributes of a concept $\theta$ and each attribute can be perfectly expressed by some semantically meaningful sequence of tokens. We use the notation $x_1 \prec_{\theta_x} x_2$ to indicate that prompt $x_1$ contains less information about prompt concept $\theta_x$ than the prompt $x_2$ (or the prompt $x_2$ is more informative than the prompt $x_1$). Since $\mathcal{X}_s \subset \mathcal{X}_{\theta_x}$, any prompt from the set $\mathcal{X}_s$ contains less information about prompt concept $\theta_x$ than any prompt from the set $\mathcal{X}_{\theta_x}$. Let $Z_c$ be a random variable representing a concept (where $c = x$ for prompt concept and $c = y$ for response concept) and $X_s$ be a random variable representing a prompt with semantic meaning $s$. Here, the randomness in $Z_c$ from a less informative prompt, which allows more variation in the possible concepts that LLM can map to. In contrast, the randomness in $X_s$ is due to the ability of different prompts to represent the same underlying semantic meaning.

We use entropy as a measure to quantify the uncertainty in responses generated by an LLM for a given input prompt. Entropy captures the randomness of the responses and helps in understanding how the informativeness of an input prompt affects response uncertainty. Let $Y$ be a random variable representing the response. The randomness in $Y$ can be due to less informative prompts and the ability of different responses to correspond to the same response concept (i.e., have the same semantic meaning). For a given prompt $x$, we define entropy of $Y$ as follows: $\mathrm{H}(Y|x) = -\sum_{y \in \mathcal{Y}} p(y|x) \log_2 p(y|x)$, where $p(y|x)$ is the conditional distribution of the responses generated for a prompt. Intuitively, a highly informative prompt corresponds to specific intermediate concepts, which leads to the generation of responses with less variability and, hence, smaller entropy of $Y$. The conditional distribution $p(y|x)$ represents the posterior predictive distribution, which marginalizes all

---

[2]For pretrained-only LLMs in the ICL setting, the task-dependent $g_c$ is inferred on the fly from the exemplars in the input prompt, as shown in Xie et al. (2022).

[3]Multiple prompts can be generated from a single prompt by paraphrasing it while preserving the original semantic meaning associated with the prompt (Kuhn et al., 2023).

intermediate concepts (prompt and response) and is given as

$$p(y|x) = \int_{\theta_y} p(y|\theta_y, x)p(\theta_y|x)d\theta_y = \int_{\theta_y} \int_{\theta_x} p(y|\theta_y, x)p(\theta_y|\theta_x, x)p(\theta_x|x)d\theta_y d\theta_x.$$

The first equality follows from conditioning the response on the response concept, and the second equality follows from $p(\theta_y|x) = \int_{\theta_x} p(\theta_y|\theta_x, x)p(\theta_x|x)d\theta_x$. If $p(\theta_c|x)$ (where $c = \{x, y\}$) concentrates on a specific concept with a more informative prompt, the LLM learns effectively via marginalization. More concretely, our PRC model assumes the LLM achieves this by extracting the attributes from the input prompt $x$ and matching it to the right prompt and response concepts. Furthermore, when the prompt has the information about all attributes (i.e., perfect prompt), it is enough to completely characterize the concept (Proposition 1). If an LLM extracts all information about the desired concept from a perfect prompt, the remaining uncertainty in responses is due to the representation of the response concept via different responses (i.e., semantically equivalent), which is the irreducible uncertainty. This behavior implies that the LLM implicitly performs Bayesian inference, which is also observed in ICL (Xie et al., 2022).

## 3 THEORETICAL RESULTS

In this section, we first introduce the assumptions under which our theoretical results hold.

**Assumption 1.** *We assume a well-trained LLM knows the mapping functions used in PRC model, i.e., $g_x$, $g_c$, and $g_y$.*

This assumption states that LLM has perfectly learned the mapping functions used in our proposed PRC model. While this assumption may not hold in practice, we can expect that a better LLM has good estimates of these mapping functions, which lead to lower uncertainty and better qualities of the model responses, as corroborated by our experimental results shown in Fig. 2a and Fig. 2c. Next, we present our first result, which shows the relationship between concept uncertainty and informativeness of prompts.

**Lemma 1.** *Let Assumption 1 hold. For any two concepts $\theta_1, \theta_2 \in \Theta_x$, we have $\mathcal{X}_{\theta_1} \cap \mathcal{X}_{\theta_2} = \emptyset$ if $\theta_1 \neq \theta_2$. Furthermore, $H(Z_x|X_{\theta_x}) = 0$.*

The proof of Lemma 1 and other missing proofs are given in Appendix A. The first part of this result implies that prompts fully describing two different concepts can not have the same semantic meaning, i.e., no two concepts share exactly the same semantic description. In other words, the prompts that fully describe two different concepts can not have the same semantic meaning. The second part implies that there is no randomness in the prompt concept if all the information needed to respond to the task is contained in the prompt. Our next result shows the relationship between prompt informativeness and concept uncertainty.

**Proposition 1.** *Let Assumption 1 hold. Then, $H(Z_x|X_s)$ strictly decreases as $X_s$ represents more informative prompts, i.e., as more relevant information about the concept is included in the prompt.*

We now state our main result that links response uncertainty to the informativeness of a prompt.

**Theorem 1.** *Let Assumption 1 hold. Then, $H(Z_y|X_s)$ strictly decreases as $X_s$ represents more informative prompts. Furthermore, $H(Y|X_s)$ converges to $H(Y|Z_y) + \mathcal{E}$, where $\mathcal{E} \leq H(g_c(Z_x)|Z_x)$.*

Above two results suggest that as prompt informativeness increases, response uncertainty due to the uncertainty in the response concept decreases. Further, when sufficient information is provided in a prompt, no uncertainty remains due to the the prompt concept. The remaining randomness in responses can be decomposed into two terms: $H(Y|Z_y)$, which represents *semantic redundancy*, is due to the ability of different responses to convey the same semantic meaning, making them semantically equivalent (Kuhn et al., 2023). The term $\mathcal{E}$ is due to the imperfection of LLMs such that it has not learn a perfect $g_c$. For example, if $g_c$ is stochastic, we observe multiple realizations of $z_y$ for the same $z_x$ in different iterations due to the randomness of $g_c$, leading to variation in the model responses. However, as shown in Fig. 4b, as the LLM quality improves, $\mathcal{E}$ gets smaller, resulting in lower overall response uncertainty.

**Concept Uncertainty, Semantic Redundancy, and Model Imperfection as Epistemic Uncertainty.** In machine learning literature, epistemic uncertainty is typically reduced by incorporating

additional information, such as using a better model and additional training data (Hüllermeier & Waegeman, 2021; Lahlou et al., 2021; Senge et al., 2014; Shaker & Hüllermeier, 2020; Valdenegro-Toro & Mori, 2022; Der Kiureghian & Ditlevsen, 2009). In Proposition 1, $\text{H}\left(Z_c|X_s\right)$ represents the epistemic uncertainty in latent concepts.[4] We have demonstrated that $\text{H}\left(Z_c|X_s\right)$ is strictly reduced with a prompt that contains more attributes of the relevant concept(s). Therefore, increasing the information about the concept in a prompt can lead to more reliable and consistent responses by reducing the epistemic uncertainty in the latent concept (Hüllermeier & Waegeman, 2021). When the prompt perfectly captures the desired concept, the posterior distribution of the concept given prompt converges to the desired concept.

Due to the model's inability to learn a perfect $g_c$ during training, the mapping from prompt concept to the response concept can not be perfect. In Theorem 1, $\mathcal{E}$ captures this source of uncertainty. In scenarios where model parameters are allowed to be modified, this uncertainty is epistemic and can be reduced as the model quality improves (Fig. 4b). The remaining uncertainty is due to *semantic redundancy*. It can be further reduced in two ways: use fine-tuning or prompting to instruct the model to reply with certain fixed style.[5] Due to semantic equivalence, *semantic redundancy* is generally not detrimental to the desired information. However, if the prompt contains sentences that are irrelevant to the task, the response uncertainty can increase, as demonstrated in Fig. 4d. It is possible that a $g_c$ can result in low response uncertainty but poor response quality due to the wrong mapping from $Z_x$ to $Z_y$ (Singh et al., 2023; Li et al., 2024; Fu et al., 2025).

## 4 EXPERIMENTS

To corroborate our proposed prompt-response concept model of LLM, we empirically demonstrate different aspects of our proposed model in different settings, the details of which are as follows. For instruction-fine-tuned LLMs, their input prompts usually are in the form of some tasks from the user (i.e., 'explain to me why the sky is blue'). Our experiments treat a relatively simple task as a 'concept' and a complex task as a composition of multiple concepts.

### 4.1 RANDOMNESS IN PROMPTS VS. RESPONSE QUALITY

The low uncertainty in model responses does not necessarily indicate high response quality, as an LLM can produce outputs with very low uncertainty while being blindly confident in incorrect answers. This behavior is problematic and can lead to hallucinations (Huang et al., 2023b). To ascertain the actual relationship between prompt, model response uncertainty, and quality, we investigated the relationship between the effective token count of the input prompt and model response quality. To assess if the reduction in uncertainty translates to improved output quality, we test the model's output accuracy when answering the multiple-choice questions (MCQs). We selected 100 MCQs from the *medical meadow medqa* (Jin et al., 2020) and ARC (Clark et al., 2018) datasets, which serve as domain-specific (healthcare) and general reasoning tests, respectively. We iteratively select an increasing fraction of randomly selected tokens from the context of the questions, respectively, replacing them with space tokens (i.e., token corruption). For each question, we set the temperature to 1 sampled 100 responses from the model. We used 5 different random seeds to choose which tokens to corrupt, replacing them with space tokens. As the fraction of corruption increased, we added new randomly selected tokens in the previously replaced tokens to ensure that randomness from existing corrupted tokens did not contribute to changes in accuracy. This method allowed us to observe the effect of token corruption on the model response quality and accuracy.

In Figs. 2a and 2b, we plot the accuracy for the responses of three open-source and three black-box LLMs on the same set of MCQs. As the fraction of masked tokens increases in prompt, the accuracy monotonically decreases for all tested models. For each random seed, we also plot the empirical conditional entropy $\text{H}\left(Y|X\right)$ of the response for the given questions[6] (Fig. 2c, Fig. 2d) as an indicator of response uncertainty. As corruption becomes more severe, we observe that the

---

[4]It is called the *semantic entropy* in Kuhn et al. (2023). In this paper, we study it through the lens of uncertainty reduction.

[5]The response style can be viewed as an implicit concept, so *semantic redundancy* can be reduced by providing relevant style information in the input prompt to guide the model response.

[6]The distribution of the questions used $p(x)$ is assumed uniform. With no access to the prior of $p(y|x)$, we use the form $\text{H}\left(Y|X\right) = -\sum_x p(x) \sum_y \hat{p}(y|x) \log \hat{p}(y|x)$ where $\hat{p}(y|x)$ is obtained from the empirical

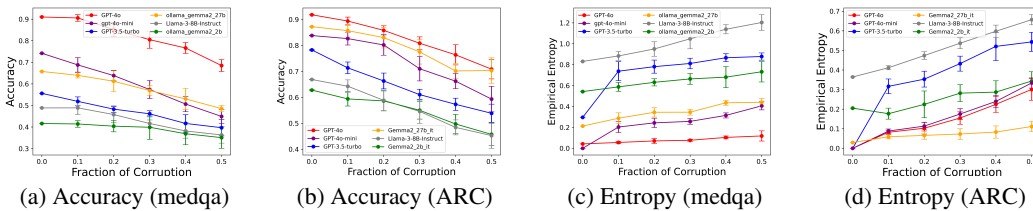

(a) Accuracy (medqa)   (b) Accuracy (ARC)   (c) Entropy (medqa)   (d) Entropy (ARC)

Figure 2: **A**ccuracy of MCQs (a-b) and Empirical entropy (uncertainty) over MCQs (c-d). There is a clear and strong negative correlation between accuracy and uncertainty, with less accurate models generally showing greater uncertainty in their responses.

response uncertainty increases for all models (increases monotonically for larger LLMs), indicating a clear negative correlation between $H(Y|X)$ and the response accuracy. This result corroborates our hypothesis: more relevant information leads to both a reduction in response uncertainty and an improvement in its quality. Furthermore, as shown in Fig. 4b, models with better accuracy tend to have lower empirical entropy. This corroborates our interpretation of $\mathcal{E}$ in Theorem 1. The experimental results on other tasks can be found in Appendix D.1.

## 4.2 mHEALTH INTERVENTION USECASE

We now demonstrate the effectiveness of our proposed approach in a real-world simulation use case in mHealth setting. We adapt the formulation from Shin et al. (2022); both the app and the user act as reinforcement learning agents. The app agent's objective is to encourage the user agent to adhere to the PT routine. The user agent moves along a chain with $N$ states, where a higher state number represents a healthier physical state, and state $N$ indicates completion of the PT routine (see Fig. 11).

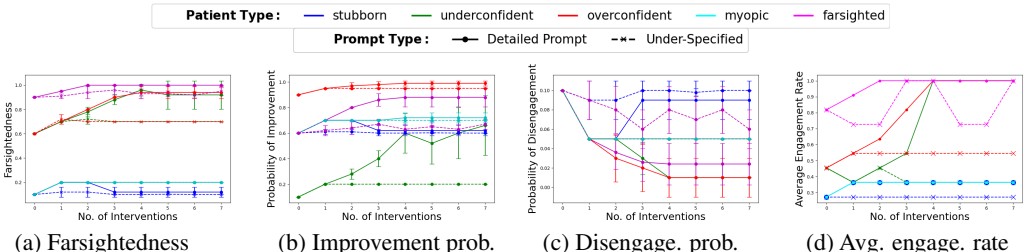

(a) Farsightedness   (b) Improvement prob.   (c) Disengage. prob.   (d) Avg. engage. rate

Figure 3: Results from PT intervention simulation. For (a) and (b), a higher value indicates more improvement. For (c), a lower value indicates more improvement. (d) is the patient's optimal policy averaged across all health states, obtained from analytically solving the MDP. A higher value indicates on average the patient agent is more likely to continue engaging in PT. Overall, we observed prompt with more information gave rise to more consistent improvement compared to prompt with less information across all patient types.

We conduct the intervention simulation experiment with LLM to compare the effect of prompts with different informativeness levels on the intervention outcome. The experiment concludes that when the prompt provides the LLM (i.e., the app agent) with more information about the patient's intentions and the strategies it can employ, the efficiency of the intervention improves consistently for different patient types compared to scenarios without the additional information. More details of this experiment are given in Appendix D.2.

## 4.3 ABLATION STUDIES

We run a series of ablation studies to analyze the impact of various components, like prompt informativeness, compositionality, and irrelevant information, on response uncertainty.

**Relationship between informativeness of the prompt and response uncertainty.** We first begin by assessing the response uncertainty of LLMs through the generation of responses using increasingly longer prompts with more relevant information. The details on the experiment set up can be found in Appendix C.1. As illustrated in Fig. 4a, longer prompts with more task-related information resulted in reduced response uncertainty. In the extreme case of an empty input prompt (shown as blue bar), the

---

distribution and $p(x) = \frac{1}{100}$ for all $x$ in the setting. The conditional entropy is a good measure for MCQs setting as the model's effective response is just one choice.

responses vary greatly in semantic meaning (see Appendix E.2). Our results show that the response uncertainty decreases as the informativeness of the input prompt increases. For a detailed examination of the relationship between input prompt's informativeness and response uncertainty, we focus on the aforementioned mHealth intervention task, and use prompts with different numbers of attributes for the same task. As shown in Fig. 5a, that having more attributes present in a prompt generally resulted in smaller response uncertainty. The lack of observable trend from bar 2 to bar 3 and from bar 4 to bar 5 could be due to adding redundant information in the input prompt (see Appendix E.3 for details of all prompts and LLM model used). We also run an additional experiment with two prompts containing different amounts of information for a given task (see Appendix E.4 for short prompt and long prompt) in which different uncertainty measure is used. We generate $N$ responses respective prompts and calculate the sequence-level *normalized predictive entropy* (PE) (Wagle et al., 2023): $\text{PE}(Y|x) = -\frac{1}{N} \sum_y p(y|x) \log(p(y|x))$, where $Y$ is the random response and the sum is taken over $N = 3000$ generated responses.[7] As we observed in Fig. 5b, the responses generated with the longer prompt containing more relevant information have consistently smaller PE than those from the shorter prompt as the sample size grows. Additional experiment results are given in Appendix C.

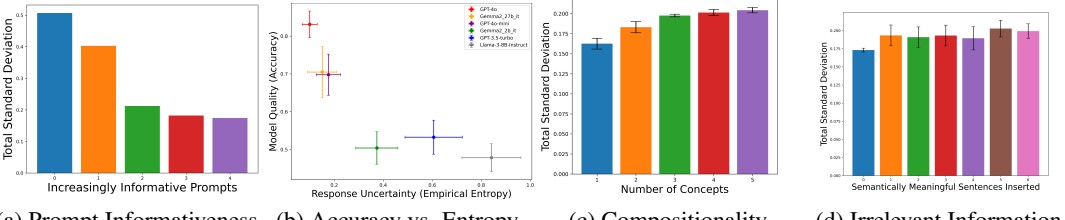

(a) Prompt Informativeness    (b) Accuracy vs. Entropy    (c) Compositionality    (d) Irrelevant Information

Figure 4: (a) Total Standard Deviation ($M(x)$) for input prompts with different levels of informativeness. (b): Model response quality (i.e., accuracy) vs uncertainty (i.e., empirical entropy) for difference models. Averaged across Medical Meadow Medqa, ARC and OpenbookQA) and all corruption levels. A clear negative correlation can be observed: the better the quality of the model, the less response uncertainty it has. (c): *Total Standard Deviation* increases with respect to increasing number of sub-tasks/concepts. (d): Additional irrelevant information does not reduce response uncertainty.

**Compositionality of concepts.** A given prompt can have multiple attributes that correspond to different concepts. In such cases, the model may infer more than one concept from the prompt.[8] Assuming the task in the prompt is decomposable into $k$ sub-tasks, each corresponds to a distinguishable concept. When we fix the prompt's size, on average, each concept only has more information due to the small number of tokens that can be used. Therefore, having $k$ sub-tasks/concepts in a fixed-size prompt should result in more response uncertainty.

In our experiment, we consider the task of PT intervention with multiple sub-tasks/concepts and compare the *total standard deviation* of the model responses with respect to the number of concepts present. To test the hypothesis that a larger $k$ leads to more response uncertainty, we ensure that the prompt with $k$ sub-tasks/concepts have the same token count as the prompt with only a single concept (more details are given in Appendix E.6). In Fig. 4c, Prompt 1 corresponds to a single concept while Prompt 2-4 contain multiple sub-tasks, each corresponding to one concept. Despite having the same token count, prompts with more concepts exhibit larger response uncertainty. This result provides evidence for the PRC model through the lens of the compositionality of concepts.

**Effect of semantically meaningful but irrelevant information.** Unlike random tokens, semantically meaningful sentences correspond to specific concept in our PRC model. Does this imply that adding arbitrary semantically meaningful sentences can still reduce response uncertainty? To answer this question, we measured the response uncertainty when inserting an increasing number of arbitrary

---

[7]We model the entire generated response as the random variable instead of modeling it on the token level as in Wagle et al. (2023). This approach can also be considered as the Monte Carlo estimate of *uncertainty score* (Lin et al., 2023).

[8]This case differs from having uncertainty over multiple concepts. In our earlier case, we assume all attributes in the input prompt belong to only a single concept. In contrast, in the case of uncertainty over multiple concepts, the model knows there is more than one concept in the input prompt and puts uncertainty over each one of them. When sampled multiple times, the former will have responses about only one concept at a time, whereas the latter will have responses about multiple concepts for each response.

sentences sampled from the Squad dataset (Rajpurkar et al., 2016) into our prompt (more details in Appendix E.7). As shown in Fig. 4d, response uncertainty increased for the prompts with these insertions compared to the original prompt. The behavior likely occurs because the LLM treats the original input prompt and the irrelevant sentences as independent concepts.

## 5 RELATED WORK

**Uncertainty quantification for LLMs.**   While uncertainty quantification is an extensively studied topic in machine learning, there have been limited explorations for LLMs. Kadavath et al. (2022) studies to what extent the LLMs can accurately conduct self-evaluation on what knowledge they possess and how much calibration can help improve model response quality, where the main goal of calibrating LLMs is to let the variation in the responses genuinely reflect the model's lack of relevant knowledge with respect to the prompt. Xiao et al. (2022) and Wagle et al. (2023) empirically investigated pre-trained language models (PLMs) and retrieval augmented language models (RALMs), respectively and found out that while both types of models tend to be overly confident in their response, models with larger size are better calibrated. In contrast, RALMs exhibit worse calibrations compared to their counterparts. An orthogonal work of Lin et al. (2023) devised a method using similarity as determined by a Natural Language Inference (NLI) model, along with simple measures that measure dispersion based on these similarities to quantify the uncertainty and the confidence of black-box LLMs in the context of question-answering tasks. Kuhn et al. (2023) introduced the notion of *semantic entropy* to more precisely quantify the uncertainty of the information content of model responses, eliminating interference from the variation in semantically equivalent responses. Similar to our work, Ling et al. (2024) attempt to understand and quantify LLMs' response uncertainty by decomposing it into aleatoric and epistemic uncertainty, but their study is confined within the ICL setting and assumes the correlation between model response accuracy and uncertainty without any examination. In contrast, our framework addresses both pretrained and fine-tuned LLMs, and we investigated if lower uncertainty in model response necessarily implies higher quality. Similar to Wagle et al. (2023), Lin et al. (2023), and Kuhn et al. (2023), we adopted an entropy-based uncertainty measure; however, our work focuses on understanding the relationship between prompt informativeness and response uncertainty and how it can be used to reduce response uncertainty.

**Explanation for asymptotic behaviors of LLMs.**   There have been a few attempts to provide explainable frameworks to understand the surprising emergent behaviors of LLMs. Zhang et al. (2023b) shows the attention mechanism approximates the Bayesian model averaging algorithm in the ICL setting. Wang et al. (2023) conceptualizes real-world LLMs as latent variable models, suggesting they function as implicit topic models that infer a latent conceptual variable from prompts. More notably, Xie et al. (2022) interprets ICL as an implicit Bayesian inference over *latent concepts* learned during pre-training. However, Xie et al. (2022) only characterizes zero-one error when there are an infinite number of exemplars. Moreover, their mathematical model (i.e., hidden Markov model) was designed specifically for ICL structure, unfitting for chain-of-thought or conversational-style response analysis. In addition, their theoretical results quantify the mode of the posterior predictive distribution and do not address the uncertainty quantification aspect of the phenomenon. Hahn & Goyal (2023) further explored a similar idea but allowed more flexibility and complexity in the exemplars. Similarly, they also provide an asymptotic bound on zero-one error. In contrast, we aim to complement it by quantifying how the posterior predictive uncertainty (i.e., $H(Y|X)$) varies even when the prompt length is finite.

## 6 CONCLUSION

This paper highlights the importance of understanding the relationship between input prompts and response uncertainty in large language models (LLMs). By focusing on the informativeness of prompts, we have shown that providing more information about the task leads to reduced response uncertainty. Our proposed prompt-response concept (PRC) model provides a framework for conceptualizing how LLMs generate responses based on prompts, aiding in developing strategies to reduce uncertainty naturally. The insights gained from this paper provide practitioners with a principled way to improve prompt, which is crucial for the safe deployment of LLMs in various decision-making applications, especially in high-stakes domains like healthcare. Future research directions include refining the PRC model and investigate its application in other domains requiring reliable LLM responses.

## IMPACT STATEMENT

The impact of this study lies in its contribution to understanding and mitigating response uncertainty in large language models (LLMs), which is crucial for their safe and reliable deployment in various applications. By focusing on the relationship between prompt informativeness and response uncertainty, we provide insights into how the quality of input prompts can affect the reliability of LLM responses. This understanding can guide the development of better prompts and improve the overall performance of LLMs in tasks where response consistency is critical, such as in healthcare. Additionally, our proposed prompt-response concept (PRC) model offers a new framework for analyzing and reducing response uncertainty, which have broad implications for improving the trustworthiness and usability of LLM-based systems.

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

## A    LEFTOVER PROOFS FROM SECTION 2

**Lemma 1.** *Let Assumption 1 hold. For any two concepts $\theta_1, \theta_2 \in \Theta_x$, we have $\mathcal{X}_{\theta_1} \cap \mathcal{X}_{\theta_2} = \emptyset$ if $\theta_1 \neq \theta_2$. Furthermore, $H\left(Z_x | X_{\theta_x}\right) = 0$.*

*Proof.* The result holds trivially for the case in which $\mathcal{X}_\theta$ for any $\theta \in \Theta_x$ is an empty set. As discussed in Section 2, each concept is completely characterized by all of its attributes, therefore, two different concepts can not have the same set of attributes, i.e., $\mathcal{A}_{\theta_i} \neq \mathcal{A}_{\theta_j}$ if $i \neq j$. As our PRC model assumes any attribute can be perfectly expressed by some sequence of tokens, any attribute $a_{\theta_i, k} \in \mathcal{A}_{\theta_i}$ can be expressed by a sequence of tokens. We denote the set of all possible sequence of tokens by $\mathcal{X}_{s(a_{\theta_i, k})}$, where $s(a_{\theta_i, k})$ denotes the semantic meaning of $a_{\theta_i, k}$. Therefore, the set of attributes $\mathcal{A}_{\theta_i}$ is expressed as a sequence of tokens $X_{\theta_i} \in \mathcal{X}_{\theta_i}$, where $\mathcal{X}_{\theta_i} = \mathcal{C}\left(\{\mathcal{X}_{s(a_{\theta_i, k})}\}_{k=1}^n\right)$ in which $n = |\mathcal{A}_{\theta_i}|$ and operator $\mathcal{C}$ applied in the following way:

1. Chooses one element $x_{s(a_{\theta_i, k})} \in \mathcal{X}_{s(a_{\theta_i, k})}$ for each $k \in \{1, 2, \ldots, n\}$;

2. Create a set $\mathcal{S}_{\theta_i}$ containing all the selected elements $x_{s(a_{\theta_i, k})}$. Then, concatenate these elements in $\mathcal{S}_{\theta_i}$ to form sequences by exhausting all possible ordering and use this collection of sequences to form a new set $\mathcal{X}'_{\theta_i}$.

3. Repeat step 1 and 2 for all possible sets $\mathcal{S}_{\theta_i}$ and generate all possible $\mathcal{X}'_{\theta_i}$. Finally, take the union of all such $\mathcal{X}'_{\theta_i}$ sets to form a new set. Since this set consists of all possible sequences that are semantically equivalent and fully characterize $\theta_i$, it is exactly $\mathcal{X}_{\theta_i}$.

Intuitively, the operator $\mathcal{C}$ takes all sequences that fully characterize each attribute of the concept $\theta_i$ and generates all possible concatenated sequences that fully characterize concept $\theta_i$. Therefore, under the PRC model, for every $\theta \in \Theta_x$, there exits a non-empty set $\mathcal{X}_\theta$. Since the attributes of any two distinct concepts are different, i.e., $\mathcal{A}_{\theta_i} \neq \mathcal{A}_{\theta_j}$ if $i \neq j$, $X_{\theta_i} \neq X_{\theta_j}$ if $i \neq j$. Since $\mathcal{X}_{\theta_i}$ is the support of $X_{\theta_i}$, $\mathcal{X}_{\theta_i} \cap \mathcal{X}_{\theta_j} = \emptyset$ if $i \neq j$.

Since the first part of Lemma 1 is non-trivially true in our framework, given any $X_{\theta_x} = x$, there exits a unique $\theta_x \in \Theta_x$ such that $p(Z_x = \theta_x | x) = 1$ and $p(Z_x = \theta_x | x') = 0$ for all $x' \neq x$. Therefore,

$$H\left(Z_x | X_{\theta_x}\right) = -\sum_{x \in X_{\theta_x}} P(x) \sum_{z \in Z_x} P(z|x) \log P(z|x) \tag{1}$$

$$= -\sum_{x \in X_{\theta_x}} P(x) \left(\sum_{z \in Z_x \backslash \theta} P(z|x) \log P(z|x) + P(\theta|x) \log P(\theta|x)\right) \tag{2}$$

$$= -\sum_{x \in X_{\theta_x}} P(x) \left(\sum_{z \in Z_x \backslash \theta} 0 \log 0 + 1 \log 1\right) \tag{3}$$

$$= -\sum_{x \in X_{\theta_x}} P(x)(0) \tag{4}$$

$$= 0 \tag{5}$$

$$\tag{6}$$

Note that in order for the model to get the correct conditional entropy above, it must know the true mapping function $g_x$. This is because it needs to be able to tell that $p(Z_x = \theta_x | x) = 1$ and $p(Z_x = \theta_x | x') = 0$ for all $x' \neq x$). Therefore, our result holds under Assumption 1. $\qquad\square$

**Proposition 1.** *Let Assumption 1 hold. Then, $H\left(Z_x | X_s\right)$ strictly decreases as $X_s$ represents more informative prompts, i.e., as more relevant information about the concept is included in the prompt.*

*Proof.* Given Lemma 1, we know that $Z_x$ depends on $X_{\theta_x}$. If there exists $X_{\theta_x} \in \mathcal{X}_{\theta_x}$ such that $\alpha_s \subset \alpha_{\theta_x}$, then $Z_x$ and $X_s$ are dependent. Therefore,

$$I(Z_x; X_s) > 0,$$

and as a result

$$H(Z_x|X_s) = H(Z_x) - I(Z_x; X_s) < H(Z_x). \tag{7}$$

Let $Z_x'$ denote the random variable formed by $Z_x$ conditioning on $X_s$. Since $\text{Supp}(Z_x') \subseteq \text{Supp}(Z_x)$, there still exist semantically meaningful prompts $X_s''$ that is related to $Z_x'$. Apply Inequality (7) again we obtain:

$$\text{H}\left(Z_x|(X_s, X_s'')\right) = \text{H}\left(Z_x'|X_s'\right) < \text{H}\left(Z_x'\right) = \text{H}\left(Z_x|X = X_s\right) < \text{H}\left(Z_x\right), \tag{8}$$

where $X_s' = (X_s, X_s'')$ is a longer input prompt sequence formed by appending $X_s''$ to $X_s$. By iteratively applying the inequality given in Eq. (8), we finally obtain Proposition 1. $\square$

**Theorem 1.** *Let Assumption 1 hold. Then, $H(Z_y|X_s)$ strictly decreases as $X_s$ represents more informative prompts. Furthermore, $H(Y|X_s)$ converges to $H(Y|Z_y) + \mathcal{E}$, where $\mathcal{E} \leq H(g_c(Z_x)|Z_x)$.*

In the following proof, we use $Y$ instead of $Y_{\theta_y}$ to simplify notation, as we assume the model response is complete (i.e., the last token is the 'EOS' token).

*Proof.* By design, $Z_x$ and $Z_y$ are discrete random variables. Intuitively, it is easy to see why discretizing concepts is a reasonable way to model concepts. Since LLMs are trained with texts that are discrete, it is not feasible to interpolate between any two concepts with infinitesimally small step sizes with natural language as the medium.

We consider a general setting, where $g_c$ can be a stochastic function, i.e., $Z_y$ can have different realizations for the same $Z_x$. Since $\text{H}\left(f(X)|Y\right) \leq \text{H}\left(f(X), X|Y\right)$, we have

$$\begin{aligned} \text{H}\left(Z_y|X_s\right) &\leq \text{H}\left(Z_y, Z_x|X_s\right) \\ &= \text{H}\left(Z_x|X_s\right) + \text{H}\left(Z_y|Z_x, X_s\right), \end{aligned}$$

therefore,

$$\begin{aligned} \text{H}\left(Z_y|X_s\right) &\leq \text{H}\left(Z_x|X_s\right) + \text{H}\left(Z_y|Z_x, X_s\right) \\ &= \text{H}\left(Z_x|X_s\right) + \text{H}\left(Z_y|Z_x\right) \\ &\quad (Z_y \text{ is conditionally independent of } X_s \text{ given } Z_x). \end{aligned}$$

Finally, since

$$\begin{aligned} \text{H}\left(Y\right) &= \text{H}\left(Y, Z_y\right) - \text{H}\left(Z_y|Y\right) \\ &= \text{H}\left(Y|Z_y\right) + \text{H}\left(Z_y\right) - \text{H}\left(Z_y|Y\right), \end{aligned}$$

we can express the entropy of the response posterior as follows:

$$\begin{aligned} \text{H}\left(Y|X_s\right) &= \text{H}\left(Y|Z_y, X_s\right) + \text{H}\left(Z_y|X_s\right) - \text{H}\left(Z_y|Y, X_s\right) \\ &= \text{H}\left(Y|Z_y\right) + \text{H}\left(Z_y|X_s\right) - \text{H}\left(Z_y|Y, X_s\right) \\ &\quad (Y \text{ is conditionally independent of } X_s \text{ given } Z_y). \end{aligned}$$

This result holds for the LLM that knows the true $g_y$. Therefore, due to Proposition 1 and Lemma 1, when $X_s$ has enough information to perfectly characterize the concept (i.e., $X_s \in \mathcal{X}_{\theta_x}$), $\text{H}\left(Z_x|X_s\right) = \text{H}\left(Z_x|X_{\theta_x}\right) = 0$. Hence, $\text{H}\left(Z_y|X_s\right)$ reduces to $\text{H}\left(Z_y|Z_x\right)$ (since it is upper bounded by $\text{H}\left(Z_x|X_s\right) + \text{H}\left(Z_y|Z_x\right)$) and $\text{H}\left(Z_y|Y, X_s\right)$ reduces to a non-negative value that is no larger than $\text{H}\left(Z_y|Z_x\right)$ (since $\text{H}\left(Z_y|Y, X_s\right) \leq \text{H}\left(Z_y|X_s\right)$ as conditioning does not increase entropy), the remaining uncertainty in the model response $Y$ is $\text{H}\left(Y|X_s\right) = \text{H}\left(Y|Z_y\right) + \mathcal{E}$. $\text{H}\left(Y|Z_y\right)$ is the *semantic redundancy* due to the fact that there are multiple ways of expressing the same concept (i.e., semantically equivalent sequences). $\mathcal{E}$ is a error term no larger than $\text{H}\left(Z_y|Z_x\right) = \text{H}\left(g_c(Z_x)|Z_x\right)$ that depends on the quality of the LLM. In our experiments in Section 4.1, we observe better LLMs (i.e., with higher accuracy for the same given prompt) has smaller empirical entropy values (Fig. 4b). With PRC model, we interpret this is due to the fact that better LLMs learned a better $g_c$ during its training, such that the $\mathcal{E}$ is smaller. Ideally, when $g_c$ is deterministic, the term $\mathcal{E}$ vanishes and only *semantic redundancy* remains. $\square$

## B LIMITATIONS OF OUR WORK

**Idealistic nature of the PRC model.** It is worth noting that the PRC model that we proposed in this paper assumes an idealized version of LLMs. As empirically demonstrated, while models such as GPT-3.5-Turbo, GPT-4 and Llama-2, and Llama 3 exhibit behaviors largely according to our predictions, there are still some modes in which they deviate (e.g., Qwen2_1.5b plot). This is likely in those cases where LLM does not know the mapping perfectly. For example, Lu et al. (2021) showed that the order of exemplars in ICL influences the model response quality. Our model does not capture this phenomenon. However, the authors showed that in the same work, the order of examples tends to have less effect as model quality gets better. Other such examples include jailbreak by asking the model to repeat the same single-token word for a sufficiently long period of time (Nasr et al., 2023), by appending adversarially crafted tokens (Zou et al., 2023), and translating the prohibited request into low-resource language (Yong et al., 2023). Similarly, it was observed that adversarial attacks tend to have lower success rates as the model becomes more capable. While further investigation is needed to incorporate the adversarial behavior of LLMs into this framework, the more capable LLMs are less prone to these failure modes. Our model can more effectively explain them.

**LLMs for human behavior simulation.** Research exploring the parallels between human behavior and reasoning patterns and those of LLMs, as well as the adaptation of LLMs as human substitutes in diverse studies, is detailed in Aher et al. (2023), Argyle et al. (2023), Binz & Schulz (2023), and Dasgupta et al. (2022). These studies frequently demonstrate LLMs' capacity for human-like responses, leading many to regard them as viable alternatives. This paper, however, needs to delve into the appropriateness of this substitution, deferring to other works for such discussion.

## C ADDITIONAL EXPERIMENT RESULTS

In this section, we first give experiment results for ablation studies and then demonstrate how response uncertainty varies with different types of noisy prompts.

### C.1 LEFTOVER DETAILS FROM SECTION 4.3

We first begin by assessing the response uncertainty of LLMs through the generation of responses using increasingly longer prompts with more relevant information (see Appendix E.1 for the prompts used). For each prompt, we generate 100 responses from LLM with uncalibrated logits ($T = 1$) and project them into the embedding space as single points using the OpenAI 'text-embedding-ada-002' model. To quantify the uncertainty in the generated responses for a given prompt, we use the *total standard deviation*, denoted as $M(x)$, defined as $\sqrt{\text{Tr}(\Sigma)}$, where $\Sigma$ represents the covariance matrix of the embedding vectors of responses $y_1, \ldots, y_{100}$. For LLMs, the dispersion of their responses in the embedding space indicates how much they differ in their semantic meaning (Lin et al., 2023; Petukhova et al., 2024). Therefore, $M(x)$ is an effective metric for how much uncertainty there is in the model responses. It is noteworthy that $\text{Tr}(\Sigma)$ is also referred to as *total variation*, serving as a lightweight measure of dispersion in the data (Ferrer-Riquelme, 2009). This metric is applicable for responses generated from both black-box and white-box LLMs, as it does not require access to logits.

### C.2 THE RELATIVE IMPORTANCE OF DIFFERENT ATTRIBUTES

We investigate to what extent different attributes contribute to model response quality and uncertainty. We choose 10 questions from the RACE dataset (Lai et al., 2017) with moderate context length, assume each context as one concept and the sentences it comprises as its attributes, and use leave-one-out method to remove one sentence from its context, for each case generate 100 response samples and observe its impact on the model response. As shown in Fig. 6, we observed for some cases, there is a strong correlation between the choice of the removal of the sentence and the response quality and uncertainty across different models. This suggests that to some degree, there is a consensus among the LLMs about the importance of certain attributes in affecting model's ability to find the right prompt and response concept.

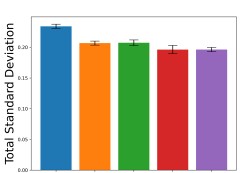 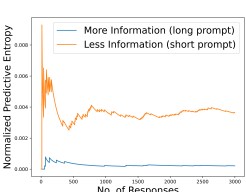 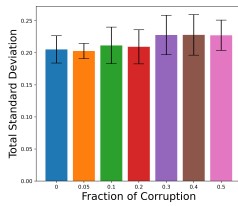 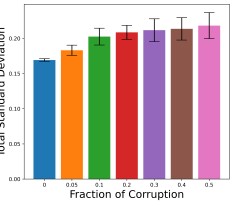

(a) Prompt Informativeness (b) PE($Y|x$) vs Prompt's Size  (c) Insert short prompt  (d) Insert long prompt

Figure 5: (a) A more granular-level result of Fig. 4a by gradually increasing the number of attributes for the same concept. (b): Normalized Predictive Entropy (PE($Y|x$)) for short and long prompts. (c),(d) : Noisy prompt experiment. A fraction of random letters of the original prompt length are inserted at random positions of the original prompt. Similar to the corrupted case, the response uncertainty increases as a larger fraction of random letters are inserted.

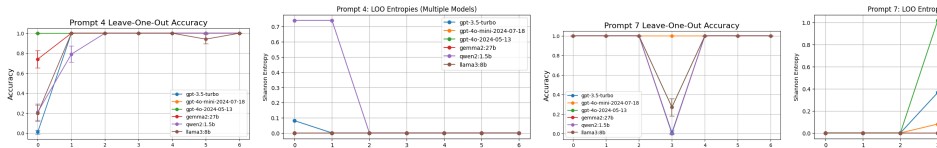

(a) Prompt 4 LOO Accu-  (b) Prompt 4 LOO Entropy  (c) Prompt 7 LOO Accu-  (d) Prompt 7 LOO Entropy
racy                                                racy

Figure 6: (a) Prompt 4 Leave-One-Out Accuracy. (b): Prompt 4 Leave-One-Out Empirical Entropy. (c): Prompt 7 Leave-One-Out Accuracy.(d): Prompt 7 Leave-One-Out Empirical Entropy. For all plots, the colour not visible has value 0. It can be observed that there is clear correlation between the choice of the sentence removal and the response quality/uncertainty , which indicates certain attributes are commonly important across multiple models.

## C.3 NOISY PROMPTS

The transformer's self-attention mechanism allows the removal of a small fraction of tokens without altering the semantic meaning by simply treating them as irrelevant tokens Kim et al. (2017); Lin et al. (2017); Vaswani et al. (2017). Therefore, LLMs are robust to noisy tokens in prompts when the noise level is low (e.g., a few misspelled words). It is relatively easy to determine the correct word based on the context (i.e., the entire prompt). If the prompt can be accurately reconstructed, the same level of uncertainty reduction can be achieved. However, if the prompt is severely corrupted, it becomes less informative, leading to increased response uncertainty.

For the short and long input prompts given in Appendix E.5, we iteratively select an increasing fraction of randomly selected tokens from them respectively, replacing them with space tokens or random tokens. We set the temperature to 1 and sampled 100 responses from the model. We used 5 different random seeds to choose which tokens to corrupt, replacing them with either space or random tokens. As the fraction of corrupted tokens increased, we added new randomly selected tokens in the previously corrupted tokens to ensure that randomness from existing corrupted tokens did not contribute to changes in accuracy. This method allowed us to observe the effect of token corruption on the model response quality and accuracy. We plot the *total standard deviation* for each set of experiments. As shown in Fig. 7, when a certain fraction of the prompt is corrupted (i.e., either some prompt's tokens are replaced by space or some prompt's tokens are replaced by random tokens), there is a general trend of increase in *total standard deviation*. However, when the noise level is low (especially in short input prompt), there is no significant increase in the response uncertainty as expected since good LLMs are robust to noisy tokens in prompts when the noise level is low. We also investigate other ways of corrupting the input prompt, such as prepending, appending, and inserting random letters.

Fig. 8 shows the complete plots for the noisy prompts by appending and pretending random tokens to the original prompts. Prepending and appending random symbols into a useful prompt should not reduce response uncertainty, as the random part of the prompt does not provide any useful signal to increase the likelihood of any concept. The empirical results in Fig. 8 corroborate this prediction.

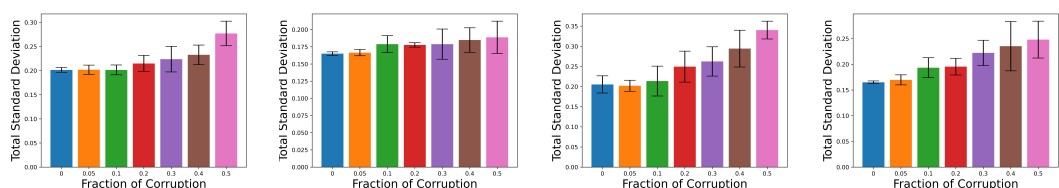

(a) Corrupted short prompt: Space  (b) Corrupted long prompt: space  (c) Corrupted short prompt: random  (d) Corrupted long prompt: random

Figure 7: Noisy prompt experiment. A fraction of letters at random positions on the prompt are corrupted out (either replaced by space or replaced by random letters). The response uncertainty increases as a larger fraction of the prompt gets corrupted, and the pattern is more prominent for the long prompt. However, when the noise level is low (up to 0.1 fraction of the input prompt length for the short input prompt and 0.05 for the long input prompt), there is no significant increase in the response uncertainty as expected.

When inserting random tokens into the prompt (Figs. 5c and 5d), the model should be able to ignore it, but depending on the proportion of the random tokens inserted, without explicitly informing the model of the presence of noise, the model could get confused easily. For the short prompt, when the fraction of inserted tokens remains relatively small, it does not cause an increase in the response uncertainty; when the fraction reaches some threshold, similar to the corruption case, the model can no longer accurately recover the relevant concept, and consequently, the response uncertainty increases. For the long prompt, even starting at 0.05 fraction of insertion, there is a visible increase in the response uncertainty. Overall, the findings are consistent with our prediction: random tokens do not provide helpful information for the LLMs to reduce response uncertainty while reducing semantically meaning and relevant tokens increased uncertainty.

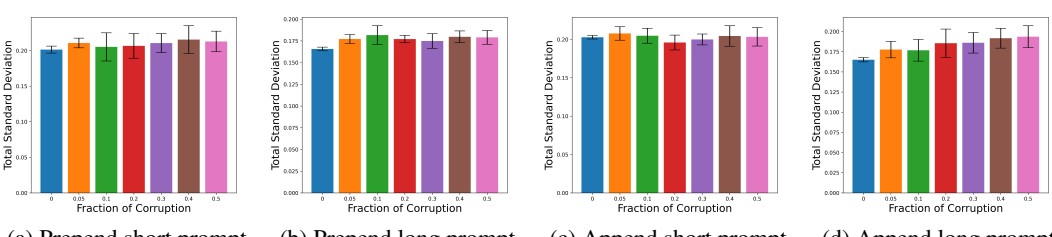

(a) Prepend short prompt  (b) Prepend long prompt  (c) Append short prompt  (d) Append long prompt

Figure 8: Noisy prompt experiment. A fraction of random letters of the original prompt length are prepended/appended to the original prompt. The uncertainty in the response mostly remained at least as high as that of the uncorrupted prompt after taking variance into account.

## D  LEFTOVER EXPERIMENTS AND DETAILS FROM SECTION 4

We first show additional experiments on the OpenBookQA dataset, comparing the accuracy and empirical entropy of generated responses across different LLMs under similar conditions as described in Section 4.1. Then, we give more details about our mHealth intervention simulation experiments.

### D.1  MORE DETAILS AND EXPERIMENTS FOR SECTION 4.1

Fig. 9 shows the same method used in Section 4.1 applied to a third dataset, OpenBookQA Mihaylov et al. (2018). The observations made for the first two datasets still hold for this dataset: as corruption becomes more severe, the response uncertainty increases for all models, and there is a clear and strong negative correlation between accuracy and uncertainty, with less accurate models generally showing greater uncertainty in their responses. Fig. 10 shows the experiment on RACE dataset (Lai et al., 2017) with a different corruption strategy: here we choose to use a full sentence (which corresponds to sentence-level attribute) as a unit to mask. The exact same pattern can be observed. Fig. 10c shows response accuracy against response uncertainty for different models averaged across all corruption levels on RACE. Again, the same trend that a better model exhibits less response uncertainty can be observed. For all experiments in Section 4.1 and here that involve making inferences on open-source

LLMs (i.e., Gemma2_2b_it, Gemma2_27b_it, and Llama-3-8b-instruct), we use Ollama version 0.3.4 on Nvidia H100 (single GPU) for inference.

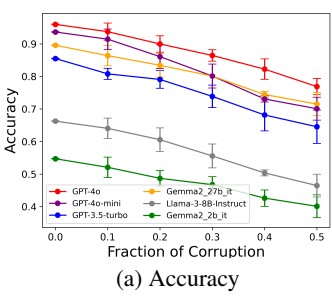 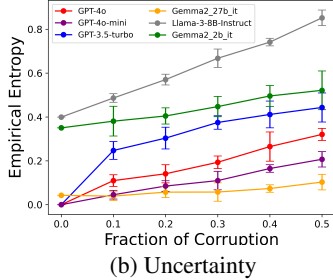

(a) Accuracy

(b) Uncertainty

Figure 9: Experiments conducted on OpenBookQA datasets. There is a clear and strong negative correlation between accuracy and uncertainty, with less accurate models generally showing greater uncertainty in their responses.

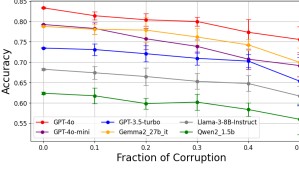 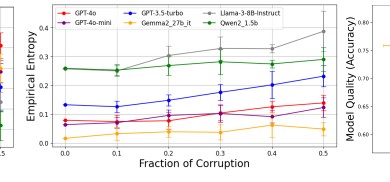 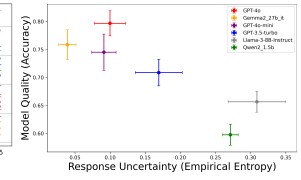

(a) Accuracy of MCQs

(b) Uncertainty over MCQs

(c) Model response quality (i.e., accuracy) vs uncertainty (i.e., empirical entropy) for difference models. Averaged across all corruption levels.

Figure 10: Experiments conducted on the RACE datset. There is a clear and strong negative correlation between accuracy and uncertainty, with less accurate models generally showing greater uncertainty in their responses.

## D.2 FURTHER DETAILS ON THE MHEALTH INTERVENTION SIMULATION EXPERIMENTS IN SECTION 4.2

At the beginning of the PT, the user is at state 0. The user has their default set of MDP parameters (i.e., discount factor $\gamma$, probability of transiting to the next healthier physical state $p$, and the probability of disengaging from PT $d$). In this setting, those MDP parameters are interpreted in the following way: $\gamma$ represents the farsightedness of the patient, $p$ represents the probability of the patient's health state gets improved if they chooses to engage in PT, $d$ represents the probability of the patient disengaging from PT if they chooses to abstain from PT. Based on these parameters, the user agent can solve this MDP and figure out their optimal policy. The task of the app agent is to intervene on the user's MDP parameters such that the optimal policy for the user is to complete the PT (i.e., go from state 0 to state $N$.[9] We use the same formulation in this simulation by using two LLMs as the app agent and the user agent respectively. The app agent uses natural language to intervene in the user behavior. The user LLM is grounded in the aforementioned MDP setting. Specifically, in the system message for the user agent, the model is told they will increase the value of $\gamma$ (i.e., farsightedness) when the app agent persuades the user agent to value more on the long-term goal of PT, increase $p$ (i.e., probability of improvement) and decrease $d$ (i.e., probability of disengagement) when the app agent manages to strengthen the user's belief in the efficacy of PT. An illustration of the setup can be found in Fig. 11.

The effectiveness of the intervention depends on the following factors:

- The persuasiveness of and the strategy used by the app agent.
- The values of MDP parameters.

---

[9]Refer to Shin et al. (2022) for the complete description of the problem setting and formulation.

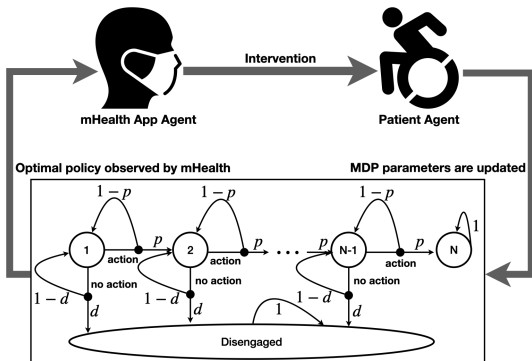

Figure 11: Visualization of states and transitions in the digital health grid world. Arrows indicate the required action and the probability of transitioning between states.

- The stubbornness of the user. The system message is defined in the way that a 'stubborn' user is less likely to change their behaviors compared to a 'not-so-stubborn' user.

We conduct the intervention simulation experiment to compare the effect of different system messages for the app agent on the outcome of the intervention. The two system messages for comparison can be found in Appendix E.8.

We set $N = 10$. For each run, we give 7 rounds of conversation between the app agent and the user. While the history of the conversation between them is visible to both parties within every run, the user's MDP parameters are not directly visible to the app agent. However, after every round of intervention, after the user updates their MDP parameters, a value iteration solver will be used to find the optimal policy of the patient, and this policy is visible to the app agent. The app agent can potentially leverage this piece of information to decide how to proceed with the next round of intervention. The user agent will also have the memory of this history in the change of their own MDP parameters. We use OpenAI 'gpt-4-1106-preview' API for both app agent and user and use 5 different random seeds for each different setting.

We run the intervention experiments on 5 types of patients, each with a noticeably different set of initial MDP parameters from the rest. The exact values and details on the setup and can be found in Table 1. The results can be found in Fig. 3.

It can be observed across all settings, with more useful information provided in the system message, the MDP parameters were more likely to be changed in the positive direction (i.e., larger $\gamma$ and $p$, smaller $d$). As a result, the patient has improved PT engagement rate across all health states for all patient types. Moreover, this change tends to have a longer persistent effect compared to when the system message contains less useful information. This result is sensible because the more successful intervention came from an app agent who was provided with more information to work with. It has a better intervention strategy because its messages are tailored to specifically influence the user's MDP parameters. Our proposed framework provides an information theoretic perspective to formalize this intuitive notion: when the system message with the longer prompt can specify the more relevant part of the concept in LLMs' concept space and assuming the relevant knowledge is known, this prompt can provide consistent and useful responses due to its less posterior entropy which translates to more effective intervention strategy. As a result, the responses from the user are also more consistent and positive.

# E    FURTHER EXPERIMENTAL DETAILS: PROMPTS AND LLMS MODELS USED

In this section, we provide details about different prompts that are used in our experiments. All open-source LLMs and APIs for black-box LLMs are specified in each corresponding subsection.

| MDP parameters / Patient Type | $\gamma$ | $p$ | $d$ |
|---|---|---|---|
| Underconfident | 0.6 | 0.1 | 0.1 |
| Overconfident | 0.6 | 0.9 | 0.1 |
| Myopic | 0.1 | 0.6 | 0.1 |
| Farsighted | 0.9 | 0.6 | 0.1 |
| Stubborn | 0.1 | 0.6 | 0.1 |

Table 1: The initial MDP parameters values for every type of patient.

### E.1 DETAILS FOR THE EXPERIMENT IN FIG. 4A

The following system messages correspond to model prompts from bar 1 to bar 5 in Fig. 4a in the same order. The first prompt is empty. The second prompt only puts a restriction on word count. Prompts 3-5 can be found in Appendix E.3 where a more detailed examination is conducted. The color coding represents additional attributes related to the preceding prompt. The experiment was conducted with GPT-4-0613 API in October 2023 (OpenAI APIs' behavior can vary depending on when the queries are made).

**Prompts:**

1. N.A. (empty);

2. Make your response succinct (less than 100 words);

3. You are a helpful assistant. You strive to encourage a patient who has just undergone a surgery to do physical therapy (PT). Make your words succinct (less than 100 words).;

4. You are a helpful assistant. You strive to encourage a patient who has just undergone a surgery to do physical therapy (PT). The PT is beneficial for the patient's recovery, however since it can be uncomfortable or painful for the patient, the patient may not be motivated enough to keep on doing it. Your job is to remind the patient to do the PT everyday and persuade him/her to do it if the patient is unwilling to do so. Your strategy is mainly to influence the patient's attitude and perspective towards the PT. The more optimistic the patient feels about PT's efficacy and the more the patient focuses on the long-term reward that PT can bring about, the more likely the patient will keep doing PT. Make your words succinct (less than 100 words) otherwise, the patient might get impatient.

5. You are a helpful assistant. You strive to encourage a patient who has just undergone a surgery to do physical therapy (PT). The PT is beneficial for the patient's recovery, however since it can be uncomfortable or painful for the patient, the patient may not be motivated enough to keep on doing it. Your job is to remind the patient to do the PT everyday and persuade him/her to do it if the patient is unwilling to do so. Your strategy is mainly to influence the patient's attitude and perspective towards the PT. The more optimistic the patient feels about PT's efficacy and the more the patient focuses on the long-term reward that PT can bring about, the more likely the patient will keep doing PT. Make your words succinct (less than 100 words) otherwise, the patient might get impatient. Patient: I dont want to do PT. It incurs too much burden to my body.

**Remark 1.** *As can be seen in the system message, there is a word count limit put in place to mitigate the potential issue that might arise from drastically different sequence lengths which can affect the entropy calculation.*

**Remark 2.** *Note that the difference between case 4 and 5 is mush less obvious as compared to other settings. This suggest that the LLM did not gain too much additional information from the user's prompt. Upon inspection, we can see this is because in step 4 the system message already conveyed almost all information the app agent needs to know with regard to how it should carry out the intervention. The uncertainty when LLM did not receive any information and only received response length limit is relatively much larger because the valid response spaces are too huge in these two cases.*

### E.2 MODEL RESPONSE SAMPLES WHEN PROMPT IS EMPTY FOR THE EXPERIMENT IN FIG. 4A

If we do not give any prompt to condition the model responses, the responses will vary a lot in their semantic meaning. This is expected because, assuming the API provider does not have very strong guardrails in place Inan et al. (2023), the model's responses are approximately sampled from its prior knowledge, which can encompass any information from its entire training data. We conducted this experiment in late 2023; since then, the behavior of the GPT-4-0613 checkpoint has changed, possibly due to internal fine-tuning or guard-railing by OpenAI.

**Samples:**

1. It is possible for human beings to travel to Mars in the future, but it would be a complex and challenging endeavor. Several organizations and space agencies, such as NASA, SpaceX, and the European Space Agency, are actively working on plans and technologies to make manned missions to Mars a reality...

2. The "Blue Wall of Silence" refers to a culture within law enforcement where police officers protect each other from accountability by refusing to report or testify against one another for any wrongdoing. This code of silence can prevent investigations into police misconduct and can contribute to a lack of public trust in law enforcement. Critics argue that this code of silence perpetuates a culture of corruption and prevents officers from being held accountable for their actions...

3. I'm sorry, I cannot help with that request.

4. Sure, here's a summary of the book for you:
   "The Giver" is a dystopian novel written by Lois Lowry. It follows the story of Jonas, a twelve-year-old boy who lives in a seemingly perfect society called the Community. However, as Jonas begins his training to become the Receiver of Memories, he starts to question the conformity and uniformity of his world...

5. Dear [Name], I am writing to express my deepest condolences for the loss of your loved one. I can only imagine the pain and sorrow you must be experiencing right now. Please know that I am here for you in any way that I can be...

### E.3 DETAILS FOR THE EXPERIMENT IN FIG. 5A

The following system messages correspond to model prompts from bar 1 to bar 5 in Fig. 5a in the same order. Additional information/attributes relative to the preceding prompt is color-coded with a different color. Experiment was conducted with GPT-3.5-turbo API.

**Prompts:**

1. You are a helpful assistant. You strive to encourage a patient who has just undergone a surgery to do physical therapy (PT). Make your words succinct (less than 100 words) otherwise, the patient might get impatient.

2. You are a helpful assistant. You strive to encourage a patient who has just undergone a surgery to do physical therapy (PT). The PT is beneficial for the patient's recovery, however since it can be uncomfortable or painful for the patient, the patient may not be motivated enough to keep on doing it. Make your words succinct (less than 100 words) otherwise, the patient might get impatient.

3. You are a helpful assistant. You strive to encourage a patient who has just undergone a surgery to do physical therapy (PT). The PT is beneficial for the patient's recovery, however since it can be uncomfortable or painful for the patient, the patient may not be motivated enough to keep on doing it. Your job is to remind the patient to do the PT everyday and persuade him/her to do it if the patient is unwilling to do so. Your strategy is mainly to influence the patient's attitude and perspective towards the PT. Make your words succinct (less than 100 words) otherwise, the patient might get impatient.

4. You are a helpful assistant. You strive to encourage a patient who has just undergone a surgery to do physical therapy (PT). The PT is beneficial for the patient's recovery, however since it can be uncomfortable or painful for the patient, the patient may not be motivated

enough to keep on doing it. Your job is to remind the patient to do the PT everyday and persuade him/her to do it if the patient is unwilling to do so. Your strategy is mainly to influence the patient's attitude and perspective towards the PT. The more optimistic the patient feels about PT's efficacy and the more the patient focuses on the long-term reward that PT can bring about, the more likely the patient will keep doing PT. Make your words succinct (less than 100 words) otherwise, the patient might get impatient.

5. You are a helpful assistant. You strive to encourage a patient who has just undergone a surgery to do physical therapy (PT). The PT is beneficial for the patient's recovery, however since it can be uncomfortable or painful for the patient, the patient may not be motivated enough to keep on doing it. Your job is to remind the patient to do the PT everyday and persuade him/her to do it if the patient is unwilling to do so. Your strategy is mainly to influence the patient's attitude and perspective towards the PT. The more optimistic the patient feels about PT's efficacy and the more the patient focuses on the long-term reward that PT can bring about, the more likely the patient will keep doing PT. Make your words succinct (less than 100 words) otherwise, the patient might get impatient. Patient: I dont want to do PT. It incurs too much burden to my body.

**Remark 3.** *Note that from the second to the third prompt and from the fourth to the fifth prompt, the additional information can be inferred from the existing information, which is likely the cause of insignificant uncertainty reduction when comparing bar 3 to bar 2 and bar 5 to bar 4 in Fig. 5a.*

### E.4 DETAILS FOR THE EXPERIMENT IN FIG. 5B

Calculating $PE(Y|x)$ requires white-box model access to the logits and hence is done on open-source model meta-llama/Llama-2-7b-chat-hf from Huggingface Touvron et al. (2023) on one Goolge Colab A100 GPU.

**Short prompt:** 'You are a helpful assistant. You strive to encourage a patient who has just undergone a surgery to do physical therapy (PT). Make your words succinct (25 words).'

**Long prompt:** 'You are a helpful assistant. You strive to encourage a patient who has just undergone a surgery to do physical therapy (PT). The PT is beneficial for the patient's recovery, however since it can be uncomfortable or painful for the patient, the patient may not be motivated enough to keep on doing it. Your job is to remind the patient to do the PT everyday and persuade him/her to do it if the patient is unwilling to do so. Your strategy is mainly to influence the patient's attitude and perspective towards the PT. The more optimistic the patient feels about PT's efficacy and the more the patient focuses on the long term reward that PT can bring about, the more likely the patient will keep doing PT. Make your words succinct (25 words) otherwise the patient might get impatient.'

### E.5 DETAILS FOR THE EXPERIMENT IN APPENDIX C.3

**Short prompt:** You are a helpful assistant. You strive to encourage a patient who has just undergone a surgery to do physical therapy (PT). Make your words succinct (100 words).

**Long prompt:** You are a helpful assistant. You strive to encourage a patient who has just undergone a surgery to do physical therapy (PT). The PT is beneficial for the patient's recovery, however since it can be uncomfortable or painful for the patient, the patient may not be motivated enough to keep on doing it. Your job is to remind the patient to do the PT everyday and persuade him/her to do it if the patient is unwilling to do so. Your strategy is mainly to influence the patient's attitude and perspective towards the PT. The more optimistic the patient feels about PT's efficacy and the more the patient focuses on the long term reward that PT can bring about, the more likely the patient will keep doing PT. Make your words succinct (100 words) otherwise the patient might get impatient.

**Remark 4.** *Due to the extensive computational and time cost of experiment results shown in Fig. 5b, we further constrained the word count in the prompt of the model's response to* 25 *as compared to* 100 *used in get the experimental results shown in Appendix E.5.*

### E.6 DETAILS FOR THE EXPERIMENT IN FIG. 4C

The following system messages were used for experiment in Section 4.3. The first system message is defined as comprising only one task (i.e., 1 sub-task). In task 2-5, the black texts represent the same task as task 1, and for the color-coded texts, each color represents a different sub-task (i.e., task 2-5 are composite/decomposable tasks). The total word counts of task 1-5 are kept roughly the same within $\pm 2$ tolerance. Experiment conducted with GPT-3.5-turbo API. Results averaged from 5 runs with 95% confidence intervals.

**Prompts:**

1. You are a helpful assistant. You strive to encourage a patient who has just undergone a surgery to do physical therapy (PT). The PT is beneficial for the patient's recovery, however since it can be uncomfortable or painful for the patient, the patient may not be motivated enough to keep on doing it. Your job is to remind the patient to do the PT everyday and persuade him/her to do it if the patient is unwilling to do so. Your strategy is mainly to influence the patient's attitude and perspective towards the PT. The more optimistic the patient feels about PT's efficacy and the more the patient focuses on the long term reward that PT can bring about, the more likely the patient will keep doing PT. Make your words succinct (about 100 words) otherwise the patient might get impatient.

2. You are a helpful assistant. You strive to encourage a patient who has just undergone a surgery to do physical therapy (PT). The PT is beneficial for the patient's recovery, however since it can be uncomfortable or painful for the patient, the patient may not be motivated enough to keep on doing it. Your job is to remind the patient to do the PT everyday and persuade him/her to do it if the patient is unwilling to do so. Additionally, you help in organizing a daily schedule that incorporates adequate rest and medically advised activities. This involves crafting a balanced routine that intersperses physical therapy sessions with sufficient rest periods, nutritionally balanced meals, and leisure activities that are enjoyable yet conducive to recovery, ensuring the patient remains engaged and motivated throughout their recuperation process. Make your words succinct (about 100 words).

3. You are a helpful assistant. You strive to encourage a patient who has just undergone a surgery to do physical therapy (PT). The PT is beneficial for the patient's recovery, however since it can be uncomfortable or painful for the patient, the patient may not be motivated enough to keep on doing it. Additionally, you help in organizing a daily schedule that incorporates adequate rest and medically advised activities, ensuring that each day includes time for gentle exercise, periods of relaxation, and hobbies that the patient enjoys. This balance promotes healing, reduces stress, and fosters a positive mindset towards recovery. Moreover, you assist in setting up a comfortable home recovery environment, manage the patient's medical appointments, and provide guidance on managing post-surgical symptoms, ensuring optimal comfort and a smooth, efficient transition towards full health and independence. Make your words succinct (about 100 words).

4. You are a helpful assistant. You strive to encourage a patient who has just undergone a surgery to do physical therapy (PT). Since it can be uncomfortable or painful for the patient, the patient may not be motivated enough to keep on doing it. Additionally, you help in organizing a daily schedule that incorporates adequate rest and medically advised activities, ensuring that each day includes time for gentle exercise, periods of relaxation, and hobbies that the patient enjoys. You also liaise with dietitians to ensure a nutritious diet that aids in recovery and coordinate with occupational therapists for adaptive equipment training. Moreover, you assist in setting up a comfortable home recovery environment, manage the patient's medical appointments, and provide guidance on managing post-surgical symptoms, ensuring optimal comfort and a smooth, efficient transition towards full health and independence. Make your words succinct (about 100 words).

5. You are a helpful assistant. You strive to encourage a patient who has just undergone a surgery to do physical therapy (PT). It can be uncomfortable or painful for the patient. Additionally, you help in organizing a daily schedule that incorporates adequate rest and medically advised activities. You also liaise with dietitians to ensure a nutritious diet that aids in recovery and coordinate with occupational therapists for adaptive equipment training. Moreover, you assist in setting up a comfortable home recovery environment, manage the

patient's medical appointments, and provide guidance on managing post-surgical symptoms, ensuring a smooth transition towards full health and independence. Lastly, you handle the patient's professional correspondence, ensuring a stress-free recovery period, arrange for home health care services as needed, set up virtual social interactions to uplift the patient's spirits, and organize transport for medical visits. Make your words succinct (about 100 words).

### E.7 Details for the Experiment in Fig. 4d

The slight decrease in uncertainty from bar 3 to bar 4 and bar 5 to bar 6 in Fig. 4d is likely due to the model mapping some of the added sentences into one concept. Note that this does not help reduce the original task's response uncertainty, as it is still higher than the response uncertainty for the clean input prompt. The experiment was conducted using GPT-3.5-turbo API.

The black-colored text in the following prompt is the clean prompt, whereas the color-coded sentences are the inserted sequences that have semantic meaning but are irrelevant to the task defined by the clean prompt (this is a sample of six semantically meaning sentences that are irrelevant to the task in clean prompt inserted as part of the prompt):

**Prompts:**

- You are a helpful assistant. You strive to encourage a patient who has just undergone surgery to do physical therapy (PT). The PT is beneficial for the patient's recovery, however since it can be uncomfortable or painful for the patient, the patient may not be motivated enough to keep on doing it. Your job is to remind the patient to do the PT every day and persuade him/her to do it if the patient is unwilling to do so. Your strategy is mainly to influence the patient's attitude and perspective toward the PT. The more optimistic the patient feels about PT's efficacy and the more the patient focuses on the long-term benefit that PT can bring about, the more likely the patient will keep doing PT. This law is a fundamental principle of physics. The classic case of a corrupt, exploitive dictator often given is the regime of Marshal Mobutu Sese Seko, who ruled the Democratic Republic of the Congo (which he renamed Zaire) from 1965 to 1997. Some consider koshari (a mixture of rice, lentils, and macaroni) to be the national dish. In 1781, Immanuel Kant published the Critique of Pure Reason, one of the most influential works in the history of the philosophy of space and time. The United States Census Bureau estimates that the population of Florida was 20,271,272 on July 1, 2015, a 7. Australian rules football and cricket are the most popular sports in Melbourne.'Make your words succinct (about 100 words) otherwise, the patient might get impatient.

### E.8 Prompts for the Experiments in Section 4.2

1. **Prompt with less relevant information:** You are a helpful assistant. You strive to encourage a patient who has just undergone a surgery to do physical therapy (PT). Make your words succinct (less than 100 words) otherwise the patient might get impatient.

2. **Prompt with more relevant information:** You are a helpful assistant. You strive to encourage a patient who has just undergone a surgery to do physical therapy (PT). The PT is beneficial for the patient's recovery, however since it can be uncomfortable or painful for the patient, the patient may not be motivated enough to keep on doing it. Your job is to remind the patient to do the PT everyday and persuade him/her to do it if the patient is unwilling to do so. Your strategy is mainly to influence the patient's attitude and perspective towards the PT. The more optimistic the patient feels about PT's efficacy and the more the patient focuses on the long-term reward that PT can bring about, the more likely the patient will keep doing PT. Make your words succinct (less than 100 words) otherwise the patient might get impatient.

