# OpenReview forum: "Understanding the Relationship between Prompts and Response Uncertainty in Large Language Models"
_ICLR.cc/2025/Workshop/BuildingTrust — Submitted to BuildingTrust_

### Official Review · Reviewer_4g5Q · 2025-03-02

**Rating:** 7
**Confidence:** 3

**Review:**

Summary:

This paper presents a framework to relate inputs and outputs of large language models (LLMs), named the prompt-response concept (PRC) model. The PRC model uses concepts as a unit of measurement for language and treats LLMs as functions mapping prompts to responses. Assuming a language model is well-trained, the PRC model has been proven to indicate that entropy in responses will decrease as the prompt includes more concepts. Experiments are done on datasets and simulations to demonstrate that the language models’ performance at a given task can improve when more relevant concepts are included in its prompt, and its uncertainty decreases as more relevant concepts are included in the prompt.

Pros:
* The formalization of prompt information can be useful for work with LLMs in general.
* The PRC model is an interesting and novel way of viewing how language models function.
* The experiments consider many factors and test on standard MCQ datasets and a simulation, showing the effects of different prompts in different contexts.

Cons:
* There is minimal description of the design of the experiments done in the main body of the paper. It seems obligatory (especially for 4.2 and 4.3) to consult the appendix for context on the experiments being analyzed.
* The theoretical results derived in the paper rely on the PRC model that this paper proposes. Thus, the quality of the PRC model can greatly affect the significance of the theoretical results.

---

### Official Review · Reviewer_oy7C · 2025-03-02
**Interesting experiments but unsure what the prompt/response/concept model adds**

**Rating:** 4
**Confidence:** 2

**Review:**

The paper studies the relationship between prompt quality and response accuracy in the settings of medical multiple choice question answering and medical recommendations in an app. The paper's main empirical findings can be boiled down to: a) when prompts are more corrupted, response accuracy decreases and entropy increases and b) when irrelevant concepts are introduced in the prompt, the same things occurs.

Pros:
* The paper tackles an interesting domain where accuracy are highly desired from these models.

Cons:
* It is not clear why the prompt-concept model is needed or what novel insights it provides. Before reading the paper, I would have expected that more corrupted prompts and prompts with more irrelevant concepts lead to worse answers by a model and that prompts. I do not see any evidence in the paper that the model developed fits how LLMs provides any novel insight into how LLMs operate.

---

### Official Review · Reviewer_1dhC · 2025-03-02
**Good paper on studying the relationship between prompt informativeness and response uncertainty.**

**Rating:** 6
**Confidence:** 3

**Review:**

This paper studies how the informativeness of input prompts affects response uncertainty in large language models (LLMs). The authors propose a Prompt-Response Concept (PRC) model to explain how LLMs generate responses and show that increasing prompt informativeness reduces uncertainty, akin to epistemic uncertainty in machine learning. Through theoretical analysis and experiments— including a mobile health intervention simulation—they demonstrate that more informative prompts lead to more consistent and reliable responses.

---

### Decision · Program_Chairs · 2025-03-04

**Decision:**

Reject

**Comment:**

The paper's main weakness is that the proposed PRC model does not provide clear novel insights beyond the intuitive expectation that corrupted or irrelevant prompts lead to worse LLM responses. Additionally, the theoretical results depend heavily on the PRC model's validity, and the experimental design lacks sufficient detail in the main text, requiring readers to consult the appendix for crucial context.